# YBX1 Indirectly Targets Heterochromatin-Repressed Inflammatory Response-Related Apoptosis Genes through Regulating CBX5 mRNA

**DOI:** 10.3390/ijms21124453

**Published:** 2020-06-23

**Authors:** Andreas Kloetgen, Sujitha Duggimpudi, Konstantin Schuschel, Kebria Hezaveh, Daniel Picard, Heiner Schaal, Marc Remke, Jan-Henning Klusmann, Arndt Borkhardt, Alice C. McHardy, Jessica I. Hoell

**Affiliations:** 1Department of Computational Biology of Infection Research, Helmholtz Centre for Infection Research, 38124 Braunschweig, Germany; alice.mchardy@helmholtz-hzi.de; 2Department of Pediatric Oncology, Hematology and Clinical Immunology, Medical Faculty, Heinrich-Heine-University, 40225 Düsseldorf, Germany; sujitha.duggimpudi@med.uni-duesseldorf.de (S.D.); Kebria.Hezaveh@uhnresearch.ca (K.H.); daniel.picard@med.uni-duesseldorf.de (D.P.); marc.remke@med.uni-duesseldorf.de (M.R.); arndt.borkhardt@med.uni-duesseldorf.de (A.B.); jessica.hoell@uk-halle.de (J.I.H.); 3Department of Pediatrics 1, Martin Luther University Halle-Wittenberg, 06120 Halle, Germany; konstantin.schuschel@uk-halle.de (K.S.); jan-henning.klusmann@uk-halle.de (J.-H.K.); 4Tumor Immunotherapy Program, Princess Margaret Cancer Centre, University Health Network, Toronto, ON M5G 2C4, Canada; 5Institute of Virology, Medical Faculty, Heinrich-Heine-University, 40225 Düsseldorf, Germany; schaal@uni-duesseldorf.de

**Keywords:** medulloblastoma, Y-box binding protein 1, PAR-CLIP, RNA-Seq, post-transcriptional gene regulation

## Abstract

Medulloblastomas arise from undifferentiated precursor cells in the cerebellum and account for about 20% of all solid brain tumors during childhood; standard therapies include radiation and chemotherapy, which oftentimes come with severe impairment of the cognitive development of the young patients. Here, we show that the posttranscriptional regulator Y-box binding protein 1 (YBX1), a DNA- and RNA-binding protein, acts as an oncogene in medulloblastomas by regulating cellular survival and apoptosis. We observed different cellular responses upon YBX1 knockdown in several medulloblastoma cell lines, with significantly altered transcription and subsequent apoptosis rates. Mechanistically, PAR-CLIP for YBX1 and integration with RNA-Seq data uncovered direct posttranscriptional control of the heterochromatin-associated gene *CBX5*; upon YBX1 knockdown and subsequent CBX5 mRNA instability, heterochromatin-regulated genes involved in inflammatory response, apoptosis and death receptor signaling were de-repressed. Thus, YBX1 acts as an oncogene in medulloblastoma through indirect transcriptional regulation of inflammatory genes regulating apoptosis and represents a promising novel therapeutic target in this tumor entity.

## 1. Introduction

Medulloblastomas mainly arise from undifferentiated neural precursor cells within the cerebellum or dorsal brainstem [1] and account for around 20% of all malignant brain tumors in children [2,3,4]. Incidence rates estimated in children (1–14 years of age) and adults (15–30 years of age) show that children are about 4.4 times more likely to be diagnosed with medulloblastomas than adults [4]. Medulloblastomas are classified according to their molecular background into four subgroups [5]: Group 1, derived from granule neuron precursors in cerebellar external granule cell layer with aberrant Sonic hedgehog signaling (SHH) [6,7]; group 2, originated from dorsal brain stem progenitors and Wnt-signaling (Wnt) driven [8]; group 3 with association to Myc overexpression in diverse precursors [9,10,11]; and group 4, which is thought to arise from deep nuclei precursors in the upper rhombic lip [12]. Overall survival rates of children diagnosed with medulloblastoma increased within the past years to 70–80% 5-year survival [13], but the outcome is dependent on the subgroups [14]. Standard therapies include radiation and adjuvant chemotherapy, which frequently result in severe side effects including impaired cognitive development especially in young patients and, thus, less invasive treatments and (ideally) targeted therapies are the focus of intense research [15,16].

The Y-box binding protein 1 (YBX1) is a DNA- and RNA-binding protein containing a highly conserved cold-shock domain, which is involved in many different cellular processes such as cell proliferation and development [17]. This domain has already been structurally solved and has an affinity to 5-metylcytosine RNA [18]. The YBX1 RNA binding motif was not consistently reproduced by different studies in the past, however, it is considered to be C/A-rich [19,20,21,22]. YBX1 is an essential transcriptional and posttranscriptional regulator controlling cellular stress responses during embryonic development [23]. When highly expressed, YBX1 acts by multimerizing at mRNAs, both increasing mRNA stability while maintaining controlled translation [24,25]. It is further thought that specific binding of YBX1 to mRNAs at the 3′UTR acts together with other proteins to stabilize the respective mRNAs [26,27,28]. Subsequently, binding of YBX1 on 3′UTR of mRNAs protects against exoribonuclease digestion, while 5′UTR binding inhibits the rate of translation by interfering with eIF4E binding [29]. Interestingly, reports have shown that YBX1 is highly expressed in different cancers, such as breast cancer, glioblastomas or myelomas [30,31,32]. Knockdown of YBX1 in glioblastoma cell lines had pronounced effects on cell survival and knockdown of YBX1 in mice led to a reduced tumor size [33,34]. 

We here show that YBX1 can also be considered as an oncogene in medulloblastomas and leads to decreased survival of various medulloblastoma cell lines upon knockdown. More specifically, we identified (through differential gene expression analysis) essential signaling pathways related to inflammation and apoptosis, which are altered in the medulloblastoma cell line UW228-3. By performing transcriptome-wide RNA binding analyses, we found that YBX1 binds mostly the 3′ UTR regions of genes, suggesting direct involvement in posttranscriptional gene regulation, including the key heterochromatin-associated protein CBX5. *CBX5* in turn is a gene silencer strongly associating with methylated H3K9 lysines, a mark for heterochromatin. Subsequently, inhibition of YBX1 indirectly leads to broad upregulation of heterochromatin repressed genes including those controlling inflammatory response genes and apoptosis signaling, ultimately leading to cell death. This renders YBX1 or its target CBX5 interesting therapeutic targets in many malignancies.

## 2. Results

### 2.1. YBX1 Is an Essential RBP for Cell Survival in Medulloblastomas

We reanalyzed microarray expression data of samples from 64 medulloblastoma patients to analyze RNA-binding proteins (RBPs) expression in patients across the four reported sub-groups [35]. Among a list of 1220 canonical RBPs represented on the microarray [36], we found that YBX1 is ubiquitously expressed in medulloblastoma patient samples across all sub-types and ranks among the top expressed RBPs on average (Figure 1a). However, there was no clustering of the medulloblastoma subgroups SHH, WNT, group 3 and group 4 based on YBX1 expression. Furthermore, to identify its role in the maintenance of cellular survival, we queried the DepMap database (https://depmap.org). This database holds information on gene essentiality (based on cell survival) across a broad spectrum of cell lines represented by the CERES score, deriving its information from genome-wide CRISPR-Cas9 library screens [37]. A lower negative CERES score determines slower cell growth or increased cell death upon knockout of the respective gene, thus rendering the respective gene as essential for cellular growth and survival. We considered all eight available medulloblastoma cell lines and found that YBX1 is among the top 20–30% quantile of essential genes among 1621 canonical RBPs targeted in the screen (Figure 1b). To confirm its essentiality in medulloblastomas, we used an siRNA pool of 30 siRNAs targeting YBX1 for the knockdown (KD) of YBX1 in medulloblastoma cell lines DAOY [38], Med8A (RRID: CVCL_M137) and UW228-3 [39]. We were able to validate the DepMap findings upon targeting YBX1 in UW228-3 but neither in DAOY nor in Med8A (Figure 1c, A1a). We confirmed that YBX1 was sufficiently targeted by the siRNA pool on both mRNA and protein levels in all three cell lines (Figure A1b–d). These results imply that YBX1 as a potential candidate for therapeutic targeting in medulloblastoma.

### 2.2. YBX1 Controls Inflammatory Response and Apoptosis Genes

We made use of our siRNA pool targeting YBX1 to perform RNA-seq before and after knockdown (KD) of YBX1 to identify its transcriptomic impact. We performed differential gene expression analysis in each cell line between negative control (NT) and YBX1 KD samples. Upon YBX1 KD, we found only 40 differentially expressed genes in DAOY, 92 DE genes in Med8A but 1,662 DE genes in UW228-3 (Figure 2a; FDR ≤ 0.05 and log2FC > 1/log2FC < −1). Since there were only minor effects on cell survival and gene expression in DAOY and Med8A, we were interested to what degree the few differentially expressed genes in DAOY and Med8A were affected in the highly responsive UW228-3 cell line. Out of the significantly upregulated genes in DAOY and Med8A, a high number of genes overlapped with those significantly upregulated in UW228-3 upon YBX1 KD (Figure 2b,c). The genes upregulated in all three cell lines are involved in inflammatory responses, such as *IFNL1*, *CXCL10*, *CXCL11* or *USP18*. 

Lastly, we were interested in potential pathways deregulated upon YBX1 KD, and whether these are shared across all three cell lines despite their variable expressional and survival responses. We performed a pathway analysis using Ingenuity Pathway Analysis (IPA) and confirmed that the reported differentially expressed genes in UW228-3 are related to apoptosis, with “death receptor signaling” being among the most significantly affected pathways (activation z-score = 3.528, *p*-value = 2.2 × 10^−11^) followed by “retinoic acid mediated apoptosis” (activation z-score = 3.357, *p*-value = 9.2 × 10^−6^) (Figure 2d). Interestingly, inflammation pathways such as “interferon signaling” (activation z-score = 3.441, *p*-value = 1.8 × 10^−12^) and “neuroinflammation signaling” (activation z-score = 5.461, *p*-value = 5.1 × 10^−9^) were also found among the top pathways (Figure 2d). The latter describes the process of how microglia control homeostasis in the central nervous system which generally serves a neuroprotective role and is described by expression of pro-inflammatory cytokines [40]. Individual upregulated genes upon YBX1 KD associated with this pathway include several apoptosis-related inflammatory genes such as *FAS*, *TNF*, several caspases (e.g., *CASP8*) and MHC class I genes.

We then sought to compare the pathway analyses across all cell lines and found that similar pathways were deregulated in DAOY and Med8A despite their modest transcriptional changes upon YBX1 KD. These were reflected in “interferon signaling” being activated in both DAOY and Med8A (Figure A2a), confirming a trend of inflammation-related genes to be equally upregulated across all three cell lines (Figure 2c, Figure A2b). Additionally, Med8A also displayed other significantly activated pathways, including “retinoic acid mediated apoptosis” as well as “neuroinflammation signaling” (Figure A2c). These results highlight the unique mechanism of YBX1 in apoptosis control via inflammatory signaling genes, but also underlines how complex the transcriptional and posttranscriptional control is across different cell lines.

### 2.3. YBX1 Targets 3′UTRs and Indirectly Regulates Heterochromatin-Repressed Apoptosis-Related Inflammatory Response Genes

In order to define direct targets from our transcriptome analysis, we performed PAR-CLIP for YBX1 to reveal its in vitro RNA-binding pattern. We found 2054 binding sites in 1476 distinct genes. Most of the binding sites were within 3′ UTRs and exons of the targeted genes (Figure 3a), as was previously reported [19,20,21,22]. The most significant binding motif for YBX1 using MEME across all 3′UTR and exon binding sites found a rather non-specific C/A rich binding pattern, which is in line with earlier findings (Figure 3b). The exact role of YBX1 in posttranscriptional control and translation varies highly among studied systems, and it further depends on a YBX1/mRNA ratio in each cell (see [29] and references therein). Since the previously reported posttranscriptional regulation of fully mature mRNAs mediated by YBX1 happens within the cytoplasm, we checked for cellular localization of YBX1 in DAOY and UW228-3. We found that YBX1 localizes primarily within the cytoplasm (Figure A3a; 72% cytoplasmic; GAPDH 85% cytoplasmic), thus indicating a potential role in mRNA stability and translation. To elucidate a potential effect of YBX1 in mRNA stability control, we were specifically interested to what degree 3′UTR binding of genes directly affects their expression upon knockdown of YBX1. We found that downregulated genes were only enriched in 3′UTR binding of YBX1 to a small degree, suggesting that these genes might have been stabilized and protected by expressed YBX1 and their mRNAs are degraded upon YBX1 KD. What was more striking though was the fact that upregulated genes upon YBX1 KD were specifically depleted of YBX1 binding, suggesting an indirect downstream effect of the knockdown increasing the expression of a vast number of genes (Figure 3c). We thus performed IPA upstream regulator analysis to identify potential genes that were responsible for the strong expressional upregulation upon YBX1 KD, which might be directly targeted by YBX1 itself. Indeed, we found that five genes were simultaneously predicted to be inhibited based on their downstream targets and were themselves downregulated in their expression (Figure 3d). Only one out of the five was also bound by YBX1 in its 3′UTR in our PAR-CLIP data, which was *CBX5* (Figure 3d,e). We also confirmed quantitative expression loss in UW228-3 with our RNA-seq (Figure A3b). We validated that the majority of known CBX5 target genes from the upstream regulator analysis were indeed upregulated upon reduced expression of CBX5 due to YBX1 KD (Figure 3f). *CBX5*, also called heterochromatin protein 1 alpha (HP1-alpha), is a DNA-binding protein that associates with heterochromatin mark Histone 3 K9 trimethylation (H3K9me3), and to some extent with H3K9 mono- and dimethylation [41,42]. It is thus an important factor in heterochromatin related gene repression, while it does not maintain histone methylation or the heterochromatin compaction but rather acts as a co-factor for transcription repression. This led us to consider specific histone modification H3K9me3 demarcating heterochromatin and CBX5 binding throughout the gene bodies and promoters of differentially expressed genes upon YBX1 KD. 

We found that specifically genes upregulated upon YBX1 KD have both the strongest coverage of H3K9me3 and CBX5, while stable and downregulated genes upon YBX1 KD have overall weaker H3K9me3 and CBX5 signal within gene bodies and at respective promoters (Figure 3g). These results suggest an indirect role of YBX1 as an oncogene signaling via histone modification associated gene *CBX5* to indirectly control transcriptional regulation of hundreds of genes. As these genes appear to be associated with an inflammatory response, this enables YBX1 to indirectly repress inflammatory response and apoptosis in medulloblastoma.

## 3. Discussion

RBPs have been found to be crucial players in posttranscriptional gene regulation in almost all kinds of tumors. We here analyzed the oncogenic effect of YBX1 in three medulloblastoma cell lines based on RNA-Seq and PAR-CLIP data, revealing important insights into its posttranscriptional functions and its potential target gene network. The high expression and essentiality of YBX1 in the tested cell lines as well as its high expression levels in medulloblastoma patients supports the hypothesis, that YBX1 also plays an important role in cell survival in medulloblastomas and likely also in this cancer acts as an oncogene as was reported for other cancers [43,44,45].

In exploring a broader pattern across the differentially expressed genes upon YBX1 KD, we found several pathways associated with inflammatory response and apoptosis signaling related to retinoic acid induced apoptosis to be activated following YBX1 KD. Several studies already showed that retinoic acid induced cell growth arrest and apoptosis is a suitable anti-tumor effect in medulloblastoma [46,47], but many cell lines including DAOY and UW228 are considered to be resistant against RA induced cell death [48,49]. Yet, different mechanisms through which retinoic acid induces apoptosis in medulloblastoma were reported, including expression of BMP2 [48] or expression of FAS and its ligand FASL [50]. Other pathways were specifically associated with (neuro-)inflammation related apoptosis. A recent report showed that YBX1 supports immune evasion in hepatocellular carcinoma cells by decreasing the release of important inflammatory cytokines such as IL10 and TGFβ [51], further underlining our findings of a negative regulation of apoptosis-related cytokines and chemokines in medulloblastoma. Importantly, despite less pronounced transcriptional effects in the other cell lines, we found similar pathways to be significantly enriched upon YBX1 KD in DAOY and Med8A. However, it is possible that other mechanisms render these cells either more resistant to YBX1 KD, or that experimental timing has had an impact on our results. On the one hand, the genome-wide CRISPR screen downloaded from the DepMap data, which included information for cell lines DAOY and UW228, compared knockdown at day 21 post transfection of gRNAs versus day 7 (i.e., 14 days difference to control time point), while our in vitro experiments have been conducted only 1–4 days post YBX1 KD. On the other hand, Med8A is derived from a group 3 medulloblastoma with MYC amplification and TP53 WT status [52]. It also expresses CBX5 at low levels (Figure A3b). In comparison, both DAOY and UW228-3 are SHH medulloblastomas with TP53 loss [52] and higher CBX5 expression. Thus, we suggest that Med8A with highly different molecular properties compared to UW228-3 is resistant to the changes upon YBX1 KD. To elucidate potential differences between responses in DAOY and UW228-3, we also found that IPA canonical pathway “neuroinflammation signaling” was inhibited in WT UW228-3 compared to WT DAOY (activation z-score = −2.48; *p*-value 8.3 × 10^−6^), however, the differentially expressed genes between WT DAOY and WT UW228-3 that are part of the pathway mostly included inflammatory cytokines such as CCL2, TLR4 and TLR5 upregulated in DAOY. This suggests that DAOY underlies a pathological neuroinflammation and might be less susceptible to YBX1 KD mediated apoptosis. This analysis also predicted CBX5 to be activated in WT UW228-3 compared to WT DAOY (activation z-score = 3.0; *p*-value 5.6 × 10^−9^), thus rendering UW228-3 more susceptible to a healthy brain-damage response and apoptosis induced by YBX1 KD signaling via CBX5. 

We also set out to define the direct impacts of YBX1 KD in medulloblastoma on posttranscriptional control. PAR-CLIP data revealed mainly targets in 3′ UTR and exons which is in line with previous publications and the known functions of YBX1 regulating translational control and mRNA stability [21]. However, this does not explain the vast upregulation of many genes in UW228-3 upon YBX1 KD, which led us to consider this to be an indirect effect. We found strong recurrent binding throughout the 3′ UTR of *CBX5*. The natural targets of CBX5 are repressed when CBX5 is expressed due to its association with repressive histone modification H3K9me3. An inflammatory gene set has been described before to be under the regulation of CBX5 [53], which becomes actively transcribed upon YBX1 KD due to degradation of then unprotected CBX5 mRNA. Because other genes than *CBX5* directly bound by YBX1 and differentially expressed upon YBX1 KD are involved in inflammatory response and apoptosis based on Gene Ontology annotations, such as *NFE2L1* and *PSMA6*, we cannot rule out that *CBX5* is not the sole driver of the cellular responses to YBX1 KD but may be accompanied by additional apoptosis driving factors.

Further evidence is given by TNFα mediated inflammation, upon which rapid removal of repressive histone marks H3K9me2/me3 and H3K27me3 leads to activation of Nf-kB and other inflammatory signaling cascades via KDM7A and UTX [54]. This is in line with our hypothesis of gene induction of heterochromatin repressed inflammatory signaling genes.

Lastly, in view of the urgent need for targeted therapies, and since YBX1 resistance-suppressing therapeutic effects have been already reported [45], this RBP represents a promising candidate for targeted therapy. Initial attempts to identify low-molecular inhibitors that compete with RNA binding have already been carried out. For instance, fisetin (a phenolic fruit/vegetable compound) was predicted to compete with RNA at the cold shock domain of YBX1, inhibiting in vitro and in vivo tumor growth of prostate cancer [55]. Therefore, it appears essential to unravel even further targetable structures related to proteins (in-)directly controlling inflammation and apoptosis in medulloblastoma.

## 4. Materials and Methods 

### 4.1. Cell Lines

Cells were maintained in DMEM (Sigma-Aldrich, St. Louis, MO, USA) including 10% fetal calf serum (Invitrogen Gibco, Carlsbad, CA, USA), 1% non-essential amino acids (Gibco), 1% penicillin/streptomycin (Gibco) and 1% sodium pyruvate (Gibco) at 37 °C and 5% CO_2_ in 175 cm² flasks. For maintenance of cell confluency, the cells were washed with phosphate buffered saline (Lonza) and detached from the flask surface using trypsin/EDTA solution (Biochrom GmbH, Berlin, Germany). For knockdown cells were seeded 24 h prior siRNA transfection at 60% confluency.

### 4.2. qRT-PCR

Total RNA was extracted using the RNeasy Mini kit (Qiagen, Venlo, Netherlands) and cDNA was synthesized using Superscript Reverse Transcriptase Kit III (Invitrogen, Carlsbad, CA, USA). qRT-PCR was performed using Power SYBR green kit (Applied Biosystems, Foster City, CA, USA). All reactions were run on an ABI 7500 real time PCR machine (Applied Biosystems) as a biological triplicate (TaqMan Assay, Hs00358903_g1, catalog number 4351372, ThermoFisher Scientific, Waltham, MA, USA). Data was acquired using the ABI SDS 2.0.1 software package. The obtained CT values were normalized per tested gene with the respective beta actin expression.

### 4.3. Western Blot and Protein Localization

The cells were washed using phosphate buffered saline (PBS, Lonza, Basel, Switzerland). After gentle rocking PBS was discarded and cells were detached by incubating with Trypsin/EDTA Solution (Biochrom) at 37 °C for 5 min and dislodged using a scraper. The mixtures were pipetted into microcentrifuge tubes and centrifuged at 450× *g* 5 for minutes. The supernatant was discarded and the cells were washed with PBS, followed by centrifugation at 450× *g* at 5 min. After removal of the supernatant the pellet was dissolved in RIPA-buffer (Sigma-Aldrich) and boiled at 95 °C for 10 min. The protein concentration was determined using the Pierce BCA Protein-Assay (ThermoFisher Scientific). Protein extracts were resolved on a 10% polyacrylamide gel, electroblotted to nitrocellulose membranes. The membranes were blocked with 5% milk solved in TBST (Sigma-Aldrich) for unspecific binding and incubated with 1:2000 diluted monoclonal rabbit anti-YBX1 (D2A11, Cell Signaling Technology, Danvers, MA, USA) at 4 °C for 24 h. The membranes were washed with TBST and incubated with 1:1000 diluted monoclonal HRP-linked anti-rabbit or anti-mouse (#7074; #7076, Cell Signaling Technology) at room temperature for 1 h, followed by an additional washing step. The membranes were analyzed using ChemiDoc (Bio-Rad, Hercules, CA, USA) and LumiGLO (Cell Signaling Technology).

For protein localization, NE-PER Nuclear and Cytoplasmic Extraction Reagents (Thermo Scientific) were used to prepare cytoplasmic and nuclear extracts. The adherent cells were harvested with Trypsin/EDTA-solution (Biochrom) and centrifuged at 450× *g* for 5 min. The pellet were washed with PBS, transferred into a microcentrifuge tube and centrifuged at 450× *g* for 5 min. The supernatant was removed and ice-cold CER I added to the pellet. The tube was incubated at 4 °C for 15 min and vortexed periodically. The mixture was supplemented with CER II, vortexed and incubated at 4 °C for one minute. After an additional centrifugation step at 16,000× *g* for 5 min, the supernatant (cytoplasmic fraction) was transferred into a clean microcentrifuge tube and stored at −20 °C. Ice-cold NER was added to the pellet and the tube was incubated at 4 °C for 40 min, interrupted by periodic vortexing. The tube was centrifuged at 16,000× *g* for 10 min, the supernatant was collected and stored at −20 °C. For protein localization analysis, the membranes were incubated with the corresponding antibodies, monoclonal rabbit anti-PARP (#9532, Cell Signaling Technology) and monoclonal mouse anti-GAPDH (Cell Signaling Technology; #97166). The amount of YBX1 and GAPDH in the fractions was determined by quantifying the bands on the membrane using ImageJ version 1.53a [56] and the macro “Band/Peak Quantification Tool”, following the respective protocol (dx.doi.org/10.17504/ protocols.io.7vghn3w). Cytoplasmic localization was calculated by dividing the cytoplasmic fraction by the sum of cytoplasmic and nuclear fraction.

### 4.4. siRNA-Mediated Knockdown

SiRNA transfections (50 nM final concentration; duration of knockdown 24–96 h; siRNA pools obtained from siTOOLs Biotech GmbH, Planegg, Germany) of DAOY, Med8A and UW228-3 cells were performed in 24-well format using Lipofectamine RNAiMAX (Invitrogen) as described by the manufacturer. Total RNA was extracted using TRIzol (Invitrogen) and further purified using the RNeasy purification kit (Qiagen). The use of a pool of siRNAs reduces off-target effects and improves on-target specificity [57].

### 4.5. Cell Viability Assay

We have performed cell proliferation assays using the CCK-8 kit (Sigma-Aldrich) as per manufacturer’s guidelines. Cells were either transfected with siRNAs targeting YBX1 or non-targeting controls as described above and subsequently seeded into 96 well plates. After addition of 10 μL of CCK-8 per well with 1 × 10^5^ cells, light absorbance at 450 nm was measured using a Tecan Photometer (Maennedorf, Switzerland).

### 4.6. RNA-Seq

cDNA libraries for subsequent sequencing were prepared according to the manufacturer’s instructions using TruSeq Total RNA sample preparation kit (Illumina) and further sequenced using Illumina HiSeq 2500 (single read, 100 cycles).

### 4.7. PAR-CLIP

PAR-CLIP from T-REx HEK293 Flp-In cells (Invitrogen) stably overexpressing FLAG/HA-tagged YBX1 was performed as described previously [58]. Briefly, cells were grown in medium supplemented with 100 µM 4SU for 12 h prior to crosslinking. After decantation of the medium, the cells were washed in PBS followed by irradiation with 0.15 J per cm2 total energy of 365nm UV light. Cells were then harvested, lysed and the cleared lysate was treated with RNase T1. Flag-HA tagged YBX1 was immunoprecipitated with an anti-FLAG M2 monoclonal antibody (Sigma) conjugated to dynabeads (Invitrogen). After a subsequent second RNase T1 digestion, beads were washed in IP wash buffer, then resuspended in dephosphorylation buffer, and lastly incubated with calf intestinal alkaline phosphatase beads. The crosslinked RNA was then radiolabelled with T4 PN kinase andwashed. Release of the protein-RNA complex from the beads was achieved by incubating the complex at 90 °C and separating on an SDS gel. The protein-RNA complex based on the size (approximately between 18 and 28 bases) was excised from the gel followed by proteinase K digestion. The remaining RNA was precipitated with phenol-chloroform extraction and ethanol precipitation. Then, standard cDNA library preparation of the 5′-32P-phosphorylated RNA was performed as follows. RNA was ligated to adapters at 3’and 5´ends followed by reverse transcription. The resulting cDNA was amplified by adapter specific primers and subjected to Illumina sequencing with an Illumina HiSeq 2500 (single end, 50 cycles).

### 4.8. Bioinformatics

#### 4.8.1. RNA-Seq

We obtained a total of 514,012,225 sequencing reads for RNA-Seq of all three cell lines prior to and after YBX1 KD. Reads were aligned with the STAR aligner [59] (version 2.7.3a; specific parameters —outFilterMismatchNoverLmax 0.05; —outFilterMultimapNmax 1) against the human reference genome GRCh38. Read counts were calculated with the STAR aligner inherent method (--quantMode GeneCounts) using gene annotations downloaded from Ensembl Genes V82. A total of 313,137,305 reads (60.92%) could be mapped to annotated genes. The obtained read counts were normalized for sequencing depth per sample and gene length by edgeR [60] (version 3.16.5) function rpkm. Differential gene expression analysis was performed using edgeR functions glmQLFit and glmQLFTest between NT and YBX1 KD in each cell line independently and results were subsequently corrected for multiple testing using the false discovery rate (FDR).

#### 4.8.2. Pathway Analysis

Pathway analysis and Upstream Regulator Analysis was performed using QIAGEN’s Ingenuity^®^ Pathway Analysis (IPA^®^, QIAGEN Redwood City, www.qiagen.com/ingenuity). We selected genes as differentially expressed for the purpose of the analysis if FDR < 0.05 and log2FC > 1.0 or log2FC < −1.0 between NT and YBX1 KD for all three cell lines.

#### 4.8.3. PAR-CLIP

We obtained a total of 136,094,488 PAR-CLIP sequencing reads. First, adapter sequences and low-quality ends were trimmed using cutadapt [61] (version 1.4.1), keeping all reads longer than 16 bases for downstream analyses. All remaining reads were aligned against GRCh38 and a transcriptome databases downloaded from Ensembl Genes V82 with the PARA-suite pipeline [62]. All 2,839,752 aligned reads were stacked into clusters with BMix in a stranded fashion (default parameters). This resulted in a total 2054 detected binding sites for YBX1 in 1476 distinct genes using homer [63] (annotatePeaks.pl function version 4.11) with hg38 annotations. We have used all binding sites falling within 3′UTRs or exons (1288 total binding sites) to conduct a motif analysis with the MEME suite [64] (parameters: zoops mode; minimum width = 8; maximum width = 20; search given strand only), and reported the top scoring motif.

#### 4.8.4. ChIP-Seq Data Integration

We downloaded pre-processed bigwig files for ChIP-Seq data for H3K9me3 (in A549 cells; accession ENCSR775TAI) and for CBX5 (in GM12878 cells; accession ENCSR372GIN) from ENCODE [65]. Non-canonical chromosome data was removed from bigwig files. Average coverage plots of H3K9me3 or CBX5 around annotated genes (Ensembl Genes V82) plus/minus 5kb that are in categories of either stably expressed, down regulated or up regulated as defined in differential expression analysis were generated with DeepTools [66] (version 2.5.7).

## Figures and Tables

**Figure 1 ijms-21-04453-f001:**
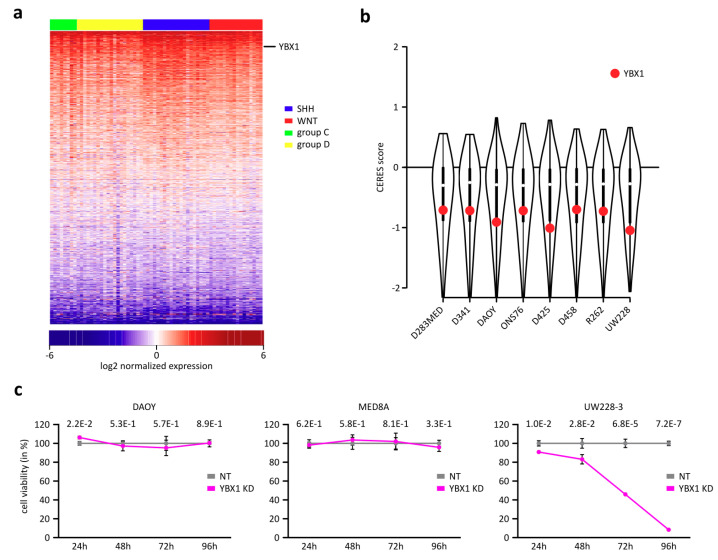
YBX1 is essential for cellular survival in medulloblastoma patients and cell lines. (**a**) Heatmap representation of a set of expression of 1220 RNA-binding proteins across medulloblastoma patients out of a total of 1827 RBPs (the remainder of RBPs was not present on the microarrays). Rows are sorted by average expression across all patients, and log2 normalized expression value is shown. (**b**) CERES score representing the essentiality of genes for survival of 483 RNA-binding proteins as in a), downloaded from DepMap for a total of eight medulloblastoma cell lines. YBX1 CERES score is highlighted. (**c**) Cell viability assay showing survival for three medulloblastoma cell lines upon YBX1 knockdown (KD) measured at different time points (YBX1 siRNA-mediated knockdown compared to control; each time point was normalized against its respective control sample).

**Figure 2 ijms-21-04453-f002:**
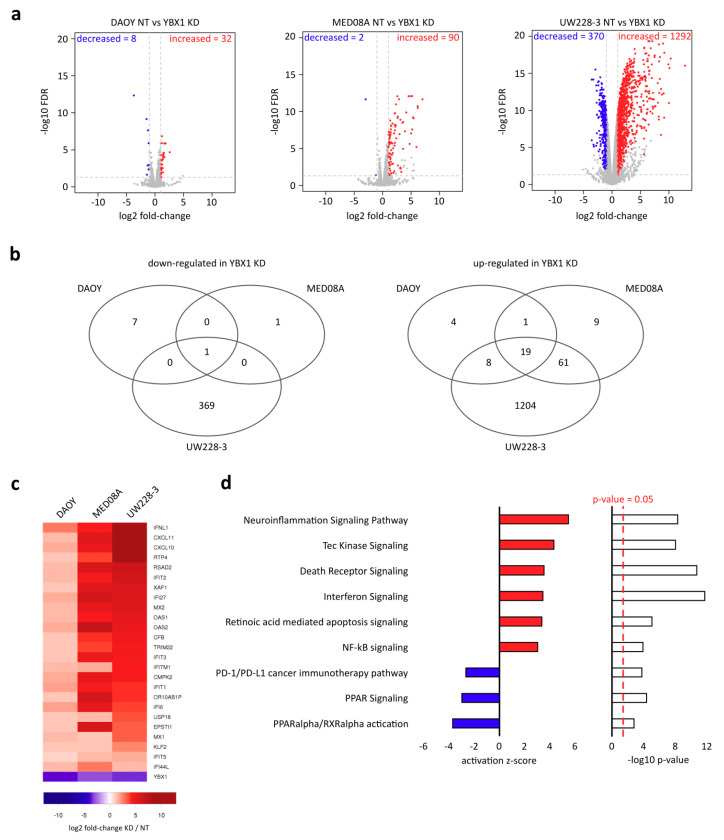
YBX1 knockdown globally induces inflammatory signaling cascades leading to apoptosis. (**a**) Volcano plot representations of differentially expressed genes in three medulloblastoma cell lines upon YBX1 KD. Differentially expressed genes are highlighted (FDR < 0.05; log2FC > 1 or log2FC < −1). (**b**) Venn diagrams for the overlap of differentially expressed genes across the three cell lines as highlighted in a). (**c**) Differential expression of commonly deregulated genes across all three cell lines as depicted in b). Heatmap shows the log2 fold-change in each cell line between YBX1 KD and NT control. (**d**) Ingenuity Pathway Analysis (IPA) was performed on all differentially expressed genes in UW228-3 as highlighted in a). Bar plots represent activation z-score and –log10 *p*-value upon YBX1 KD of the canonical pathway analysis of IPA.

**Figure 3 ijms-21-04453-f003:**
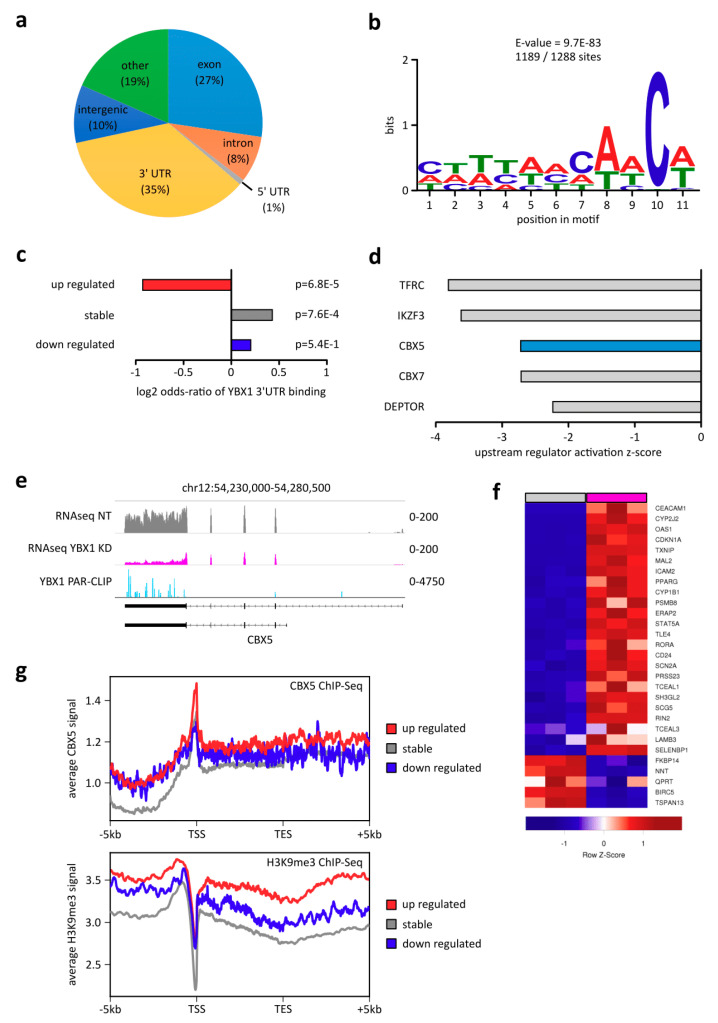
YBX1 binding to CBX5 mRNA indirectly affects repression of inflammatory pathway genes. (**a**) YBX1 PAR-CLIP performed in HEK293 cells reveals binding pattern of YBX1 using all Bmix detected binding sites within gene annotations. Annotation was performed with homer annotatePeaks.pl. (**b**) Motif analysis of YBX1 binding sites within 3′UTRs and exons (1288 total binding sites) was performed with the MEME suite, and the most significant motif is shown. (**c**) Enrichment analysis of 3′UTR/exon binding of YBX1 within differentially regulated genes in UW228-3 upon YBX1 KD. Enrichment was performed in R using Fisher’s exact test (fisher.test), and log2 odds ratio and *p*-value are reported. (**d**) IPA upstream regulator analysis was performed with all differentially expressed genes in UW228-3. Only genes with an activation z-score of < −1.96, expression log2 fold-change < −1.0 and expression FDR < 0.05 are shown. Blue highlight further depicts genes with 3′UTR/exon binding of YBX1 defined by YBX1 PAR-CLIP. (**e**) Tracks represent RNA-Seq of UW228-3 (normalized FPKM reads) and PAR-CLIP of YBX1 (normalized CPM reads) around the *CBX5* gene. Tracks were created with IGV. (**f**) Heatmap of all known downstream targets of CBX5 as reported by IPA upstream regulator analysis. (**g**) ChIP-Seq signal of CBX5 binding in GM12878 cells (upper) and H3K9me3 in A549 cells (lower) within and +/−5kb around genes reported to be differentially expressed in UW228-3. Average signal across all regions per category was created with DeepTools.

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
