# Peer review of "YBX1 Indirectly Targets Heterochromatin-Repressed Inflammatory Response-Related Apoptosis Genes through Regulating CBX5 mRNA"

_ijms, 2020, doi:10.3390/ijms21124453_

Round 1

Reviewer 1 Report

Overall, I consider the manuscript topic to be really interesting from a board perspective, novel, of value to the scientific literature and suitable for the journal scope. In general the study is presented well.

There are queries below that should be addressed:

  1. Have you tried doing immunostaining on these cells showing endogenous/overexpression of YBX1 protein? And may be doing coimmunostaining showing YBX1 and CBX5 interaction? It would be nice to show the change in expression of these proteins when you knockdown the YBX1 using siRNA.
  2. What about the off target effects of siRNA? To show the specificity can you rescue it by cotransfecting siRNA resistant YBX1?
  3. How do you confirm your YBX1 knockdown efficiency? Is it possible that you see minor effects on cell survival and gene expression in DAOY ad Med8a as compared to UW228-3 because of difference in knockdown efficiency?
  4. Did you find other targets of YBX1 other than CBX5 that may also be responsible for controlling inflammatory response and apoptosis signaling?

Author Response

Please find below our point-by-point response to the reviewers comments:

Reviewer 1:

Overall, I consider the manuscript topic to be really interesting from a board perspective, novel, of value to the scientific literature and suitable for the journal scope. In general the study is presented well.

Response: First of all, we would like to thank the reviewer for her/his support of our work.

There are queries below that should be addressed:

1.1: Have you tried doing immunostaining on these cells showing endogenous/overexpression of YBX1 protein? And may be doing coimmunostaining showing YBX1 and CBX5 interaction? It would be nice to show the change in expression of these proteins when you knockdown the YBX1 using siRNA.

Response: We thank the reviewer for the comment. Unfortunately, we have not performed immunostaining of YBX1 protein or CBX5 mRNA/protein. Also, given the short time frame of 10 days for revising our manuscript, we were not able to order YBX1 antibody for immunostaining (our current one is not suitable for this application) nor CBX5 anti-sense RNA oligonucleotides for RNA immunostaining of the CBX5 mRNA to show co-localization of the protein-mRNA interaction. However, in order to at least give a better understanding of the CBX5 mRNA instability mediated by YBX1 knockdown, we have added the quantification of CBX5 expression from our RNA-seq data before and after YBX1 KD in Supplementary Figure 3.

1.2: What about the off target effects of siRNA? To show the specificity can you rescue it by cotransfecting siRNA resistant YBX1?

Response: We thank the reviewer for this critical question. We have ordered a commercial set of 30 siRNAs (siTOOLs Biotech), each targeting a different position in the YBX1 mRNA. Under a broad assumption of each cell taking up only a single of these 30 siRNAs in a uniform manner, off-target effects of individual siRNAs are marginal in the cell population and can thus be ignored. The advantage of such “siPools” was explained and proven in Hannus et al. 2014. Furthermore, this approach makes it almost impossible to generate siRNA-resistant cell lines, because resistant cells will only be resistant to a single (or a few at maximum) siRNAs, and a new transfection would most likely result in YBX1 knockdown in a cell resistant to one/few different siRNAs. We have now outlined the approach better in the methods section (page 10) as follows:

“The use of a pool of siRNAs reduces off-target effects and improves on-target specificity (Hannus et al. 2014).”

1.3: How do you confirm your YBX1 knockdown efficiency? Is it possible that you see minor effects on cell survival and gene expression in DAOY ad Med8a as compared to UW228-3 because of difference in knockdown efficiency?

Response: We thank the reviewer for her/his question. We have now confirmed YBX1 knockdown efficiency additionally by qRT-PCR and have added the results to Supplementary Figure 1. The knockdown efficiency of YBX1 as well as its basal expression in WT cell lines is comparable across all three cell lines tested. We believe that the differences in cell survival can be attributed to other factors, as we explain in more detail in response to reviewer 2 comment #2.1.

1.4: Did you find other targets of YBX1 other than CBX5 that may also be responsible for controlling inflammatory response and apoptosis signaling?

Response: First of all, we would like to note that our analysis leading to CBX5 as a key target of YBX1 was very stringent and hypothesis-based. We specifically searched for genes that are significantly downregulated, their predicted activity based on IPA upstream regulator analysis was significantly inhibited and bound by YBX1 in the 3’ UTR. This analysis revealed only CBX5 to meet all before-mentioned criteria (Figure 2d). However, in order to satisfy an alternative and less stringent approach, we annotated all direct YBX1 targets (exon or UTR only) based on PAR-CLIP data using Gene Ontology Terms. We filtered for all GO terms related to “inflammation” or “apoptosis”, and validated their differential expression. Only two out of 25 unique genes were significantly upregulated: NFE2L1 and PSMA6 (see Rebuttal Table 1). Despite both genes being involved in inflammatory responses, neither seems to be able to explain the broad upregulation of genes upon YBX1 KD as CBX5 does. We have added a note of caution though in the discussion as follows (page 9):

“Because other genes than CBX5 directly bound by YBX1 and differentially expressed upon YBX1 KD are involved in inflammatory response and apoptosis based on Gene Ontology annotations, such as NFE2L1 and PSMA6, we cannot rule out that CBX5 is not the sole driver of the cellular responses to YBX1 KD but may be accompanied by additional apoptosis driving factors.”.

Rebuttal Table 1: Genes directly bound by YBX1 within UTR or exon based on PAR-CLIP data. Annotation of genes were performed using GeneOntology Terms (“biological processes” only) downloaded via Ensembl BioMart, and further divided into being related to inflammation or apoptosis. Gene differential expression information is derived from UW228-3 upon YBX1 KD. All non-highlighted genes are not significantly differentially expressed.

YBX1 targets related to inflammation

YBX1 targets related to apoptosis

ADCY1

AKT1

AKT1

DLC1

APPL1

DNM1L

BRD4

MTRNR2L4

DHX33

MTRNR2L7

FEM1A

PAK2

IL5RA

TAOK1

KPNA6

LDLR

MIF

NFE2L1

NR1D2

PDCD4

PNMA1

PSMA6

RAC1

RB1

RBPJ

STAT3

upregulated (FDR < 0.05 & log2FC > 1.0)

downregulated (FDR < 0.05 & log2FC < -1.0)

Reviewer 2 Report

The manuscript highlights an interesting aspect of the regulatory pathways involved in the development of medulloblastoma. The manuscript is well-written and easy to follow. The techniques used are comprehensive and the experimental setup is adequate. However, I have some concerns, mainly regarding the cell survival after YBX1 KD and the interpretation of the results.

  1. Out of the three cell lines used, only one showed marked alteriations in vitality upon YBX1 KD. Although the authors make an attempt to explain these differences, I feel that the conclusion that YBX1 is a valid therapeutic target needs more experimental support. At least a detailed physiological explanation of the observed survival differences is in place here: some  information on the genetic background of these cell lines, what subgroups of medulloblastoma they represent and other differences that might be responsible for the lack of effect.
  2. Since the authors attribute the outcome of YBX1 KD to its indirect effect on CBX5 mRNA stability,  it would be interesting to know whether the basal CBX5 levels are similar in the tested (and differently reactive) cell lines.
  3. The exact involvement of CBX5 and YBX1 in cancer development based on the presented results could be outlined in a more detailed way as in the current form there seems to be a slight contradiction. YBX1 is overexpressed in many cancer cells, but, according to the manuscript, it is it's knockdown and the subsequent destabilization of the CBX5 mRNA that induces inflammatory pathways. Some more details on how exactly the YBX1 overexpression contributes to cancer cell survival would make the interpretation of the results clearer.
  4. A minor comment: there's a mistake in the list of abbreviation, the explanation for PBS is missing and the list is shifted from there.

Author Response

Please find below our point-by-point response to the reviewers comments: 

Reviewer 2:

The manuscript highlights an interesting aspect of the regulatory pathways involved in the development of medulloblastoma. The manuscript is well-written and easy to follow. The techniques used are comprehensive and the experimental setup is adequate. However, I have some concerns, mainly regarding the cell survival after YBX1 KD and the interpretation of the results.

Response: At this point, we would like to thank the reviewer for the support and for raising critical questions about our interpretation of the results.

2.1: Out of the three cell lines used, only one showed marked alterations in vitality upon YBX1 KD. Although the authors make an attempt to explain these differences, I feel that the conclusion that YBX1 is a valid therapeutic target needs more experimental support. At least a detailed physiological explanation of the observed survival differences is in place here: some information on the genetic background of these cell lines, what subgroups of medulloblastoma they represent and other differences that might be responsible for the lack of effect.

Response: We thank the reviewer for this critical question. We have now added some explanation to the discussion, especially about the molecular background of all three cell lines as nicely summarized in Ivanov et al. 2016. In particular, Med8a was derived from a patient with a different medulloblastoma subgroup tumor as well as TP53 WT compared to DAOY and UW228-3. It additionally displays MYC amplification. Furthermore, Med8a does not express CBX5 at levels as DAOY or UW228-3 do (see new Supplementary Figure 3b). We thus conclude that these are likely explanations for the lack of Med8a to respond to YBX1 KD, and have updated our discussion accordingly (page 8):

“On the other hand, MED8A is derived from a group 3 medulloblastoma with MYC amplification and TP53 WT status (Ivanov et al. 2016). It also expresses CBX5 at low levels (Figure A3b). In comparison, both DAOY and UW228-3 are SHH medulloblastomas with TP53 loss and higher CBX5 expression. Thus, we suggest that MED8A with highly different molecular properties compared to UW228-3 is resistant to YBX1 KD.”

However, DAOY and UW228-3 appear highly similar based on these facts, so we further investigated molecular differences between the cell lines as suggested by the reviewer. We conducted a differential expression analysis between DAOY and UW228-3 WT cells. This revealed an inhibited “neuroinflammatory signaling” in UW228-3 (activation z-score = -2.48; p-value = 8.3E-6). On a closer look at the genes involved in this pathway in the WT comparison, we did not find apoptosis related genes such as caspases, FAS and others but indeed neuroinflammatory cytokines/chemokines such as CCL2 (https://pubmed.ncbi.nlm.nih.gov/20029451/ ), TLR4 (https://www.sciencedirect.com/science/article/abs/pii/S0165572814000319?via%3Dihub) (https://pubmed.ncbi.nlm.nih.gov/30680070/) or TLR5 to be upregulated in DAOY. This potentially renders UW228-3 more susceptible to inflammation-induced apoptosis than DAOY. Indeed, when checking genes from UW228-3 WT vs YBX1 KD associated with the same IPA canonical pathway “neuroinflammation signaling”, we found many inflammatory related apoptosis-genes such as FAS, TNF (https://www.nature.com/articles/s41467-018-04092-0 shows TNFa translationally regulated by YBX1), several caspases (e.g. CASP8 https://pubmed.ncbi.nlm.nih.gov/14688019/ ), MHC class I genes and others. Thus, we conclude that the term “neuroinflammation signaling” from the IPA canonical pathway analysis is quite misleading as it contains genes both related to chronic neuroinflammation causing several disorders and potentially cancers by disrupting the blood-brain barrier as perceived in the literature as well as apoptosis related genes that are part of a healthy brain-damage response.

We now clarified these confusions by several edits throughout the manuscript, including changing the title to “YBX1 indirectly targets heterochromatin-repressed inflammatory response related apoptosis genes through regulating CBX5 mRNA” and highlighting apoptosis-related genes from the “neuroinflammation signaling” IPA pathway at the appropriate position in the manuscript (page 3).

2.2: Since the authors attribute the outcome of YBX1 KD to its indirect effect on CBX5 mRNA stability,  it would be interesting to know whether the basal CBX5 levels are similar in the tested (and differently reactive) cell lines.

Response: As mentioned above in response to comment #1.1 and #2.1, we have now added CBX5 expression quantifications before and after YBX1 knockdown across all three cell lines to Supplementary Figure 3. We found that Med8a expresses CBX5 at low levels compared to higher levels of expression in DAOY and UW228-3. We would also like to highlight the IPA upstream regulator results predicting CBX5 to be inhibited in DAOY compared to UW228-3 (activation z-score = 3.0; p-value 5.6E-9), which is now part of our discussion (page 8):

“This analysis also predicted CBX5 to be activated in WT UW228-3 compared to WT DAOY (activation z-score = 3.0; p-value 5.6E-9), thus rendering UW228-3 more susceptible to a healthy brain-damage response and apoptosis induced by YBX1 KD signaling via CBX5.”

2.3: The exact involvement of CBX5 and YBX1 in cancer development based on the presented results (review in frontiers: Why Be One Protein When You Can Affect Many? The Multiple Roles of YB-1 in Lung Cancer and Mesothelioma)

could be outlined in a more detailed way as in the current form there seems to be a slight contradiction. YBX1 is overexpressed in many cancer cells, but, according to the manuscript, it is it's knockdown and the subsequent destabilization of the CBX5 mRNA (https://www.jbc.org/content/278/37/35516 ) that induces inflammatory pathways. Some more details on how exactly the YBX1 overexpression contributes to cancer cell survival would make the interpretation of the results clearer.

Response: We thank the reviewer for this comment and agree that our wording was misleading. We have elaborated on the issue of contradiction and cancer involvement in response to comment #2.1. We have also added the following to the discussion to emphasize why YBX1 might be overexpressed is cancers (page 8):

“Other pathways were specifically associated with (neuro-)inflammation related apoptosis. A recent report showed that YBX1 supports immune evasion in hepatocellular carcinoma cells by decreasing the release of important inflammatory cytokines such as IL10 and TGFβ (Tao et al., 2019), further underlining our findings of a negative regulation of apoptosis-related cytokines and chemokines in medulloblastoma.”

2.4: A minor comment: there's a mistake in the list of abbreviation, the explanation for PBS is missing and the list is shifted from there.

Response: We thank the reviewer for this very cautious observation and have corrected the list of abbreviations.

Round 2

Reviewer 1 Report

None

Reviewer 2 Report

Thank you for considering my suggestions, the clarity and comprehensiveness of the manuscript has improved a lot.